# Interdisciplinary Cooperation in Technical, Medical, and Social Sciences: A Focus on Creating Accessibility

**DOI:** 10.3390/ijerph192416669

**Published:** 2022-12-12

**Authors:** Dominika Zawadzka, Natalia Ratajczak-Szponik, Bożena Ostrowska

**Affiliations:** 1Zakład Terapii Zajęciowej, Wydział Fizjoterapii, Akademia Wychowania Fizycznego, 51-612 Wrocław, Poland; 2Innovation and Business Center, Wrocław University of Science and Technology, 50-370 Wrocław, Poland

**Keywords:** accessibility, universal design, interdisciplinarity, interprofessional learning

## Abstract

Accessibility and Universal Design (UD) is an area of professional interest for architects and occupational therapists, but college curricula rarely include both broader and collaborative education in this area. This article presents the experience of the inter-university, interdisciplinary project “Joint Architecture Initiative” (JAI), with the participation of students from the University of Science and Technology, University of Health and Sport Science, and Academy of Fine Arts in Wroclaw (Poland). The JAI project is a response of the university community of Wroclaw to the social-urban campaign “Life Without Barriers” and the needs of residents—the elderly and people with disabilities—for adaptation and modification of housing. The paper presents the theoretical background of the problem, the stages of implementation of the JAI project from the perspective of the model—human–environment–occupation—the tasks of project team members, and the justification for the need to create interdisciplinary teams from the area of technical and health sciences, with particular emphasis on occupational therapy practice (OTP).

## 1. Introduction

The accessibility of public and communal areas is extremely important in every indivdual’s life. Establishing an accessible environment, promoting health, and enabling people to participate in a community come into prominence in the context of elderly and disabled people [1,2]. The promulgation of moderate and advanced disability in the world is estimated in 15% of the general population and 50% among people over 60 years. These people experience many everyday problems relevant to physical functioning and barriers in their surroundings such as houses, flats, or the community [3,4]. Basic activities are limited by the lack of proper spacious conditions and make humans dependent on other individuals [5]. The research shows that disability may be decreased by adapting to the environment [6]. Few studies prove that the concept of universal design and home modifications improves functioning and decreases dependence [7] and has an impact on general sensation [8], participation [9], and higher self-assessment of health [10]. Noninclusive public housing, environmental barriers, and accessibility problems in the home or in the close neighborhood limit participation and do not help with health promotion or enhancing activities. Creating an accessible environment for everyone requires a comprehensive approach to the matter [1,11].

Interdisciplinary cooperation among specialists from various scientific fields—technical, medical, and social—significantly increases the effectiveness of accessibility and universal design projects. As a result of this cooperation, specific, more extended, and updated knowledge is created. The effect of this knowledge may be the development and implementation of a particular project. The exploration of a problem by different professionals, often presenting unique perspectives, enables different approaches to a specific subject matter and the exchange of experiences and competencies [11]. An example of such multidisciplinary cooperation between technical, medical, and social sciences to create accessibility is the Joint Architectural Initiative (JAI) project.

### 1.1. Accessibility and Universal Design (UD)

Accessibility for people with disabilities is essential to being active. Following the UN Convention and Article 9: “*To enable persons with disabilities to live independently and participate fully in all aspects of life, States Parties shall take appropriate measures to ensure to persons with disabilities access, on an equal basis with others, to the physical environment, to transportation, to information and communications, including information and communications technologies and systems, and to other facilities and services open or provided to the public, both in urban and in rural areas. These measures, which shall include the identification and elimination of obstacles and barriers to accessibility, shall apply to, inter alia: (a) buildings, roads, transportation and other indoor and outdoor facilities, including schools, housing, medical facilities and workplaces*” [12].

Therefore, the direction of change has been outlined: People with disabilities, on par with those without disabilities, deserve a dignified, independent life, the possibility of self-determination, and full participation in all its spheres. All of this is based on full accessibility and equality.

It is vital to note the position of the European Commission on accessibility issues: *“Accessibility and the elimination of barriers are at the heart of the European Commission’s Strategy, which has identified these aspects as the first and basic area of action for the next decade”* [13].

The concept above is consistent with the principle of universal design, which is closely related to the ideas of functionality and accessibility. Under Art. 2 of the UN Convention on the Rights of Persons with Disabilities (Journal of Laws of 2012, item 1169 and of 2018, item 1217), *“Universal design means the design of products, environments, programs, and services to be usable by all people, to the greatest extent possible, without the need for adaptation or specialized design. Universal design shall not exclude assistive devices for particular groups of persons with disabilities where this is needed”* [14].

Within the International Classification of Functioning, Disability, and Health (ICF), disability is a term used to describe the combination of the negative aspects of functional loss, activity limitation, and participation restriction. Disability is understood as a result of the interaction of impairments with personal, social, and environmental characteristics [15]. Knowledge of these factors is essential in the universal design process.

Universal Design is currently an important element in shaping health policy and promoting accessibility in the built environment [16]. The United Nations covenant about disabled people’s rights was signed by Poland in 2006 and was ratified in 2012. Thereby, Poland is obligated to bring into effect the standards of the proceeding including the universal design and accessibility [17].

### 1.2. A Homogeneous Team and an Interdisciplinary Team

The differences between a homogeneous and interdisciplinary team should be considered [18]. The first team type is a team of professionals working in similar specialties with equivalent knowledge, experience, and a similar way of thinking. It is possible to look at the problem unilaterally in homogeneous teams. In such a group, specialists may be exposed to shortcomings related to, for example, the lack of brainstorming on the subject matter, which makes it impossible to generate ideas that could contribute to developing many areas of a project. In homogeneous project teams, it is often observed that work is method-focused, and such an approach is more important than focusing on a specific audience or audiences. Such a procedure may result in omitting important issues related to the welfare of an entity for which a given project is being developed. After analyzing the work in homogeneous project teams, you can see that expanding the team with a group of specialists from various fields or areas is necessary. Working in a group of professionals from various fields enables a broad spectrum of project analysis [11]. The combination of many scientific/professional disciplines provides the possibility of a comprehensive analysis of a problem and its optimal solution.

An interdisciplinary team refers to the cooperation of various professional groups, which, using the knowledge from their fields of specialization, are better able to approach solving a problem. An increased amount of knowledge, different approaches to the subject matter, greater familiarity with the nomenclature, and learning from each other are the main advantages associated with the emergence of multidisciplinarity in design and implementation teams [11].

Methods in multi-professional teams should be adapted to a specific client, investor, or beneficiary, who must be placed in the center of a project, and not the other way around [11,16,19,20]. The recipient of the target concept should be the most important link on which the project team must focus. Moreover, an investor should also be part of the organizational and executive team. They can assume the role of an observer who can answer any questions troubling the specialists. As a result, a project, at the center of which is and should be a person, will be created following the *client-centered practice* principle [20]. The family is also part of the therapeutic team [19]. A holistic approach to the problem supported by knowledge and experience is a significant aspect of the design and executive team. Increased professional diversity means different personalities that influence the processes of creating and implementing a project.

In recent years, there has been a tendency to change the approach to design. More importance is attached to creating projects for an individual user, placing the user at the center of the project. It focuses on the social inclusion of this person, which is consistent with the philosophy of occupational therapy due to the principle of client-centered practice. A person and their needs are at the center, and justice and equality are highly desirable values [16,21].

In Poland, due to demographic variations, higher interest is seen in social and health projects relevant to establishing “no barriers environment” This phenomenon can be observed both in the case of nationwide and local projects [22].

The environment should be friendly, functional, accessible, tailored, and, as far as possible, designed in a universal manner. It is well known that many people with disabilities face barriers daily. These obstacles are often found in their immediate surroundings, such as their apartment or house. The lack of adequately designed spatial conditions essentially limits the basic activities of such a user and significantly inhibits their everyday functioning [4,10].

### 1.3. The Role of Occupational Therapy in UD

There is a synergy between the occupational therapy practice framework, the PEO model, and the UD. The application of universal design principles is one method used by occupational therapy practitioners in environmental modification [20,21]. Occupational therapists are knowledgeable about diseases, disabilities, aging, and how these processes affect functioning, and more specifically, how improving the home and community environment through UD and accessibility improves functional ability outcomes [23,24].

Designers do not have this knowledge or direct access to the perspective of people with disabilities for whom they design, so OTPs can share their knowledge about human disability in the context of meeting needs [25].

### 1.4. The Effort of Teaching and Learning

There are few research studies showing the capabilities of using UD in the execution of common educational projects by students of occupational therapy, architecture, or industrial design [26]. One of these research studies considered an inter-vocational project named DOT (designed + OT), exclusive to people with multiple sclerosis [27]. The design team was made up of seven students at the second level of industrial design, one student at the second level of occupational therapy, and two young adults with multiple sclerosis. The goal was the common creation of “planning capabilities’’ with the aim of improving the quality of the life among people with multiple sclerosis. In another research study, the combination of simulative tasks of reality and virtual reality was used by students at the first level of architecture and students at the third level of occupational therapy. They participated in the semestral course of UD and its use for disabled people [28]. However, the literature suggests weak cooperation between both sides as an obstacle for UD. These studies do not exclude the participation of OT in interdisciplinary project teams [29].

However, the literature suggests weak cooperation between both sides as an obstacle to the bigger picture of UD [29]. Those research studies confirm the value of implementing occupational therapy in planning projects at the beginning of the training, despite many different logistic inconveniences.

## 2. Materials and Methods

The aim of the research was to present the initiative of academic education and teaching in interdisciplinary teams composed of architects, occupational therapists, physiotherapists, and designers, according to the concept of universal design and accessibility, as in the inter-university Joint Architectural Initiative project.

### 2.1. Inter-University Joint Architectural Initiative Project (JAI)

The JAI project started in 2008 as a communal-technical-health enterprise facing challenges regarding “life with no barriers”. Above all, this innovative undertaking aims to support people with various disabilities and their family members in obtaining funds from the PFRON program, aimed at removing architectural, technical, and communication barriers. JAI participants receive complete construction documentation and a cost estimate for the implementation of a given project, which is required to obtain financial support in institutions such as Social Welfare Centers [22].

In 2017, the JAI project entered a new stage. The team of tutors and students of Architecture from the University of Science and Technology was joined by students and tutors from the Department of Occupational Therapy at the Academy of Physical Education. The initiative gained a new dimension and became an interdisciplinary undertaking focusing on the various needs of the investor, client, and person with disabilities. In 2018, students of Interior Architecture and Design from the Academy of Fine Arts in Wrocław were also invited to participate in the initiative. Since then, there has been an even more significant modernization of the project. Individuals work in interdisciplinary, inter-faculty, and inter-university teams consisting of tutors and students, architects, occupational therapists, physiotherapists, interior designers, and designers.

The tutors of JAI set students big educational, workshop, and planning challenges.

The diversity of the participants, both in the form of students with different professions and the variety within the investor group, prompted tutors to modify their educational approach and workshops themselves. The discussions within the teams, which often use a different language, had to be modified too. The project was also enriched with new elements that were able to expand the competencies of students participating and sensitize them to the needs of people with various psycho-physical limitations. The tutors decided to meet the challenges and extended the existing workshop convention of the Joint Architectural Initiative project.

In addition to the “City Action”, relying on taking public transportation and moving around a city in a wheelchair they, introduced the “Hour of Darkness”, during which the participants could empathize with people with sensory disorders (people with sight and hearing impairments). Furthermore, it was decided to organize a “Unique break with a meal”, during which young adepts could feel what people, for example, after upper limb amputation, feel because both their upper limbs were bandaged and they had to prepare a meal.

In order to show students the entire production cycle of rehabilitation equipment, teachers organized a trip to Vermeiren (a company producing rehabilitation equipment on a large, global scale), where the students learned about various types of equipment supporting the movement of people with physical disabilities.

The next step was the visit to the “House of Resourcefulness” at the Association of Friends of Children and People with Disabilities.

The participants also participated in a conference where various issues related to accessibility and creating a barrier-free environment were presented. During the meeting, we hosted distinguished guests from the world of regional and national politics, associations, and foundations, as well as people interested in changing the existing environment. The guests of honor were, of course, investors and their families.

In the workshop part of the JAI project, the creativity of young designers aimed to identify and diagnose problems and then propose possible changes in the existing space. At the end of the workshop, students prepared a preliminary project cost estimate and project documentation [22]. In Table 1, the schedule of the WIA project is presented.

### 2.2. The Role of Specific Professions in Inter-Academic Project “JAI”

Working in an interdisciplinary team allows architects to delegate tasks unrelated to their specialization and transfer them to students studying in different professions who are substantively prepared to cope with them and have appropriate research tools.

Future architects were primarily responsible for the project’s entire technical and drawing aspects. For this purpose, they made inventories, created design assumptions, and developed all the visual elements so that the design was correct and attractive in terms of reception. It is important to emphasize that the architectural work also differed slightly depending on the specific design team. In the case of private investors, students had to work on an existing apartment, which they modified, adapting it to the needs of a given investor. In the group dealing with the project of the center for the Potrafię Pomóc Foundation, they created the concept of a new facility. Therefore, specific designs were different and focused on different problems of adapting a given environment to the needs of people with disabilities.

At the initial stage, architects adapting apartments to the needs of private investors were involved in making an inventory. They had to know the premises’ condition and the barriers the user encountered. Creating an inventory included taking measurements, photos, and technical drawings, as well as learning about the habits of moving around the space and/or using devices and equipment. After finishing the inventory, the designers, in cooperation with other interdisciplinary team members, created two or three concepts for solving the encountered problem. Then the projects were jointly consulted with substantive supervisors and investors, after which the best option to adapt an apartment was selected. In the next phase, the chosen concept was refined in terms of functionality and visuals. After the design works were completed, the future architects were asked to prepare design documentation (drawing and descriptive parts), the adaptation cost estimate, and design charts. Documentation prepared in this way allowed the investor to apply for funding from PFRON funds for the elimination of architectural barriers.

The future occupational therapists conducted in-depth interviews based on the COMP questionnaire. They familiarized themselves with the dysfunction of a given patient/client and identified limitations and occupational problems. They analyzed the situation in an environmental and social context. They examined the client’s daily schedule, behavior patterns, tasks, and activities. As a result of analyzing the needs of a given investor, the therapists could signal to the rest of the team which part of the apartment area should be modified. They suggested introducing specific changes to the architectural plan and sometimes gave specific guidelines for designing tailored and individualized equipment supporting a given activity or occupation. In addition, they co-designed an ergonomic structure for a particular person with a disability.

Another group of students involved in developing the concept within JAI was physiotherapists. This group also interviewed the patient from a slightly different angle compared to the occupational therapists. Based on an analysis of the investor’s condition, they advised architects on, for example, where to install specific supporting equipment. Physiotherapists proposed solutions for equipping the space with specialized rehabilitation equipment such as pillows or orthopedic mattresses. Teaching the patient (in cooperation with occupational therapists) to maintain an ergonomic body posture was also of great importance.

An additional competence of physiotherapists was to educate the patient and their family on the matter of exercises that would support their functioning at a later time.

The 8th edition of the JAI was also attended by Interior Architecture and Design students from the Academy of Fine Arts. The Interior Design students learn how to arrange, furnish, and decorate interiors, including colors, lighting, and the selection of textures. They also deal with ergonomics [15]. Students from this field took part in the project of the center for the Potrafię Pomóc Foundation. Their tasks in interdisciplinary teams resulted in being very important.

### 2.3. Participants/Disabled People/Investors

During the 8th edition of JAI, four projects related to adapting the places of residence of private investors and a conceptual project of a Diagnostic and Therapeutic Center for Rare Diseases with the House of Resourcefulness for people with Prader–Willi syndrome for the Potrafię Pomóc Foundation were created. The investors included a person with dementia, a person with multiple sclerosis, a person in a wheelchair, and a blind person. The implementation of the architectural concept of the “Potrafię Pomóc” Foundation Complex for Exceptional Children from scratch was a unique project in this edition. According to the investors’ assumptions, this facility was to include a rehabilitation and educational center for a day stay, including a therapeutic kindergarten and revalidation rooms, a space for rehabilitation camps, and an intervention center, as well as an element intended to be a phenomenon on a national scale—the House of Resourcefulness for people with Prader–Willi syndrome.

In the present article, we focus on presenting two projects of private investors. Control and content-related supervision of the project were the responsibility of professors from the college.

The first project was performed for an 86-year-old woman with dementia (MMSE-22-pkt) and a physical disability. The team was made up of two students of architecture, two students of industrial design, two students of first-level occupational therapy, and one student of physiotherapy.

The task of the project group was to increase the safety and functioning of the investor in her place of residence and grant her as much independence as possible for the longest run possible.

The interview, ADL (4/6 points), IADL (14/24 points), MMSE (23/30 points), and COPM (Table 2) showed that the investor had difficulties carrying out daily hygiene activities and using the bathtub.

Architecture students focused on this aspect and designed a bathroom tailored to the needs of a person with mobility difficulties (Figure 1). They proposed a general change in the room’s space arrangement and equipped it with the necessary handles. They also suggested removing the bathtub in favor of a large shower equipped with a seat.

However, the most severe problem limiting her independence in this patient’s case turned out to be dementia and, often, problems with naming objects and understanding their purpose. To help the investor in everyday functioning and simultaneously enable her to exercise her memory and limit the aggravation of issues related to memory loss, the idea was to attach stickers to individual items and elements of interior design. This system was based on the investor’s passion for solving crosswords and was developed by occupational therapists and a designer. Their development considered the colors, size, and font type. The students not only designed but also made these stickers and handed them over to the investor (Figure 2).

The second project was completed for a 39-year-old man with Duchenne muscular dystrophy. The team was made up of two students of architecture, one student of industrial design, two students of first-level occupational therapy, and two students of physiotherapy.

The investor, with significantly limited mobility, used a wheelchair and required the constant help of an assistant. The only convenience in the man’s house was the platform in the staircase, a compressive mattress, and risers (seldom used).

The project envisaged adjusting the room and bathroom to suit his needs. The architects focused on this aspect. In the sanitary part, they suggested changing the arrangement of the equipment in the room and introducing new sanitary facilities to adjust to the needs of a disabled person. In the investor’s room, the architecture students suggested changes in the appearance of the room and the replacement of the existing equipment (a new bed and desk). An essential part of the project was the introduction of a ceiling lift between the room and bathroom, which would enable a safe transition between the rooms and individual pieces of equipment (Figure 3).

Important elements of the project for this investor were specially designed and adapted devices and furniture (Figure 4). While working on this task, occupational therapists and a designer played a significant role. They proposed, among other things, the following changes:The bathroom—mounting additional support at the handle, a washbasin with a special shape ensuring closer “contact” of the user with the device, and a shaver holder allowing for independent and comfortable shaving while only moving the head and wrist.A new desk with a retractable top and a shape, ensuring the greatest possible movement scope for the user while working in the room (Figure 5).

It was a very innovative project and entirely completed. It involved strict architectural elements and furniture and tiny details. Moreover, the physiotherapists presented the patient with various exercises to improve whole-body general motor activity. The broad scope of the project was possible only thanks to the work of the interdisciplinary team.

## 3. Summary

The roles of experts and specialists from various scientific and professional fields are changing significantly. More and more often—to achieve better results—interdisciplinary teams are created. Professionalism, understanding of specific roles, good cooperation, and communication in a team guarantee a high level of service. An example of an interdisciplinary approach to improving citizens’ quality of life is a change in thinking about broadly understood accessibility, which should take into account a wide spectrum of needs of various users [25,26,27,28,29].

The Joint Architectural Initiative project is a combination of interdisciplinarity in order to achieve accessibility. This initiative brings together participants from various, completely different scientific fields. Thanks to their involvement, the quality of the project increased, and it has become an innovative undertaking on a national scale. Project teams created within the JAI project conduct comprehensive analyses of problems and explore the best possible solution, which would not be possible in homogeneous teams. The presence of students from the fields of Architecture, Occupational Therapy, Physiotherapy, Interior Architecture, and Design is a fantastic benefit for the quality of projects. It also increases the value of the final result. Thus, the final product works better for the investor, client, or patient.

The JAI project was positively evaluated by students and investors. Students showed readiness and willingness to learn in interdisciplinary teams. Initially, the architects reported more negative impressions of interprofessional collaboration compared with the occupational therapy students. Understanding of the interdisciplinary cooperation in the JIA project, along with participation in subsequent workshops, increased compared to before the project.

There was evidence that housing modifications protected against risk of disability outcomes. Studies on reducing disability in aging populations need to consider the role of housing modifications as key interventions to promote healthy aging. In research on housing interventions to reduce the risk of disability, there is seldom any mention of interdisciplinary teams regarding accessibility [2,3,4,5,6,7,8].

The literature suggests poor collaboration between stakeholders as a barrier to the wider field of UD [29]. UD is not a high priority in architecture education and design communities [30]. OCPs can be valuable partners in interprofessional projects. OTs have knowledge about the physical, sensory, and cognitive aspects of disability and the relationship between the person, environment, and occupation.

However, despite these findings, OTs should continue to advocate for their role in creating UD spaces and products.

## Figures and Tables

**Figure 1 ijerph-19-16669-f001:**
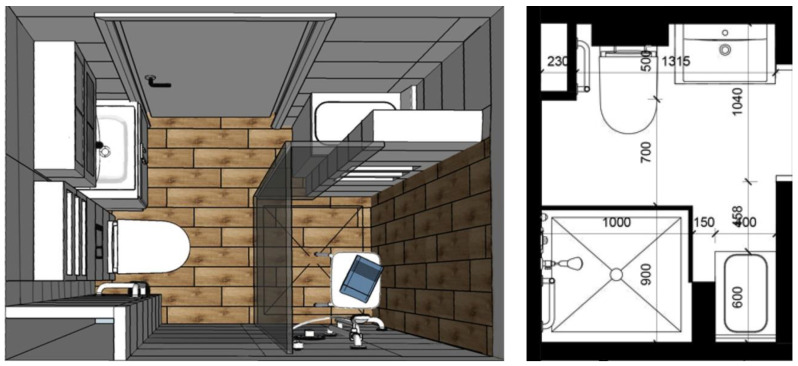
Bathroom arrangement project for an investor with dementia (picture: Students participating in the 8th edition of JAI).

**Figure 2 ijerph-19-16669-f002:**
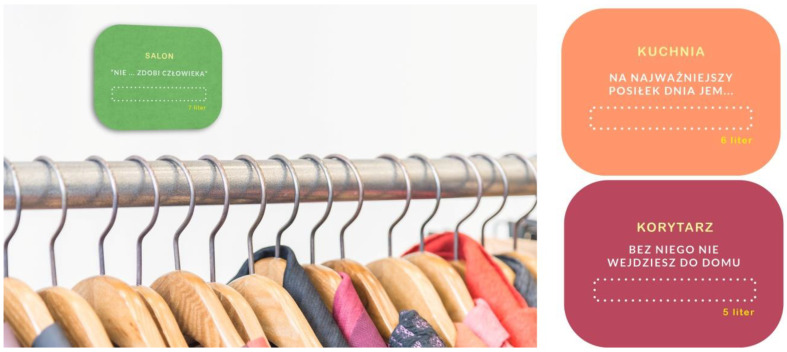
Design of stickers for an investor with dementia (picture: Students participating in the 8th edition of JAI).

**Figure 3 ijerph-19-16669-f003:**
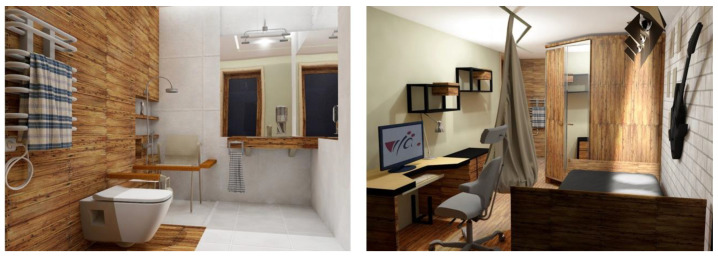
Interior IL.03. Design project for an investor with muscle atrophy—visualization (picture: students participating in the 8th edition of JAI).

**Figure 4 ijerph-19-16669-f004:**
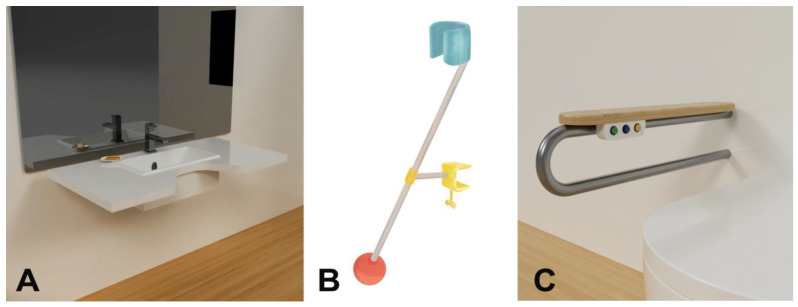
Design of individual devices for an investor with muscle atrophy: (**A**)—washbasin, (**B**)—shaver holder, (**C**)—support for the handle (picture: Students participating in the 8th edition of JAI).

**Figure 5 ijerph-19-16669-f005:**
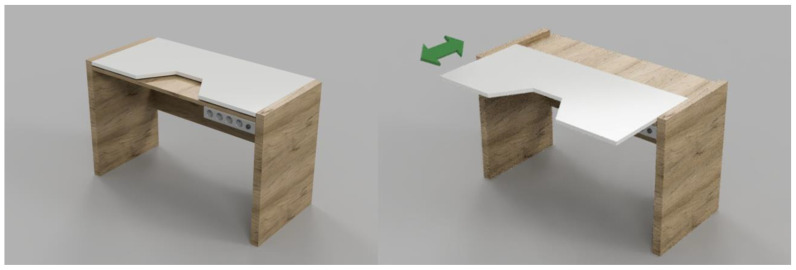
Design of a desk for an investor with muscle atrophy (picture: Students participating in the 8th edition of JAI).

**Table 1 ijerph-19-16669-t001:** The schedule of the Joint Architectural Initiative project.

Topic	The Project Recipients
Initiate The inauguration—VIII edicion Joint Architectural Initiative	Disabled people and their families, students, non-governal organisation.
The students’ recruitment	Students
The meeting with the students, students’ declarations	Students
The investors recruitment	Disabled people and their families, non-governal organisation.
Home visits Data collection: observations by means of standardized instruments study and specific questions—ADL, COPM *, occupation-based activity, characteristic of the person according to PEO (conceptual model of the occupation) and ICF Inventory	Disabled people and their families, students
Visitation: Vermeiren (a company producing rehabilitation), “Training House”	Disabled people, students
Training how to fill in the application of grant. Liquidation of architectural barriers and in communication. Submitting of declaration of participation in the project made by disabled people	Disabled people, students
The City Action: Wrocław	Students
Participation in the methodologist conference 13th Integration Forum “Living without barriers”	Disabled people and their families, students, non-governal organisation
Workshops: The presentation of the stocktaking and the first project ideas	Students
Workshops: The planning projects	Students
The meeting with investors/presentations/education	Disabled people and their families, students
Workshops: lecture of estimate and making estimates	Students
Close/Conclude 8 edycion JIA Closing ceremony	Disabled people and their families, students, non-governal organisation.

* COPM—Identification of Occupational Performance Issues.

**Table 2 ijerph-19-16669-t002:** Assessment of activity performance and satisfaction level for COMP (86-year-old woman).

A Problematic Activity	Importance	Execution	Satisfaction
Bathing—getting into the bathtub; and getting out of the bathtub	8 7	6 4	6 6

## Data Availability

The study did not report any data.

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
