# Peer review of "Interdisciplinary Cooperation in Technical, Medical, and Social Sciences: A Focus on Creating Accessibility"

_ijerph, 2022, doi:10.3390/ijerph192416669_

Round 1

Reviewer 1 Report

Thank you for your work, it is interesting and needed. I believe you need to look at two concepts for this to be complete. The first is to explain what a disability is within the context of your project. For example, a person with muscle atrophy may not be seen as disabled in a context where cognition is the measure, for example any academic setting, yet here they have a disability. Context and disability go together and your paper would be strengthened by your explaining how it is conceptualized here. 

Also, in the example for the person with dementia, putting stickers with only text is probably not as effective as say stickers with pictures of how to use the objects from their point of view. Adding in an author with expertise in dementia, education, etc might be helpful.

Author Response

Thank you very much for your valuable comments, which allowed us to improve our work.

The correction of the text is in the chapter: Accessibility and universal design:

Within the International Classification of Functioning, Disability and Health (ICF), disability is a term used to describe the combination of the negative aspects of  functional loss, activity limitation, and participation restriction. Disability is understood as a result of interaction of impairments with personal, social and environmental characteristics. Knowledge of these factors is essential in the universal design process.

Reviewer 2 Report

- the topics presented in the manuscript ¨accessebility¨ and ¨interdisplinary teams¨ are not new, as they already exist in the previous literature with more details and insight on the procedural design solution planning and technical and hierarchical roles of the interdisciplinary team members in the process. the authors should exhibit more significant data on the process of accessibility design and interdisciplinary team workflow. 

- the introduction section does not exhibit a literature review of previous accessibility design projects or interdisciplinary design teams in such projects. 

- there is a lack of detailed and scientific categorization of the disabilities, only categorizing them into main headlines without sufficient criteria, as each of these titles includes multiple categories and each of these categories requires specific and scientifically formed criteria of the design solution.  ¨Based on their members' different competencies, multidisciplinary teams have planned optimal solutions to improve the living conditions of their investors – people with various types of disabilities: physical, intellectual, visual, and cognitive.¨ the authors must include a matrix of categorizing the disabilities on more solid criteria with references and exhibiting how the interdisciplinary team was addressing each category in analysis and design procedures. 

- the authors describe in a very limited and haphazard way the methodology of multiple experiments that they have conducted in the project ¨The project was also enriched with new elements that were to expand the competencies of students participating in it and sensitize them to the needs of people with various psycho-physical limitations. The tutors decided to meet the challenges and extended the existing workshop convention of the Joint Architectural Initiative project. In addition to the "City Action," they introduced the "Hour of Darkness," during which the participants could empathize with people with sensory disorders (people with sight and hearing impairments). Also, it was decided to organize a "Unique break with a meal," during which young adepts could feel what people, for example, after upper limb amputation, feel because both their upper limbs were bandaged, and they had to prepare a meal.¨ the authors must categorize these experiments addressing the different categories of disabilities and describe more scientifically the methods of the experiment as the used probes, the number of participants, data samples, statistical analysis, and exhibit the results in diagrams or charts or any sort of measurable quantitative data rather than the redundant shallow description provided in the manuscript that gives the reader no clue about the significance of the study. 

- again the authors do not scientifically describe the site visits that they performed, as they should have mentioned the various sections of these institutions that they visited with their participants and which probes the students were introduced to. ¨In order to show students the entire production cycle of rehabilitation equipment, teachers organized a trip to the Vermeiren (a company producing rehabilitation equipment on a large, global scale), in Trzebnica, where the students learn about various types of equipment supporting the movement of people with physical disabilities. The teachers went a step further in the field of innovation in the last edition of the JAI project. They organized a trip for the students to the Center in Milicz and the "House of Resourcefulness" run by the Milicz Association of Friends of Children and People with Disabilities. The participants also participated in a conference where various issues related to accessibility and creating a barrier-free environment were presented

- the entire manuscript uses a storytelling method of exhibiting events more than exhibiting in measurable scientific data as matrices, diagrams, tables, or figures. in this paragraph the authors describe a design solution without even using a figure to exhibit at least the plan of these architectural designs, the authors must use a matrix or table for each design solution that they mention exhibiting at least the plan and architectural program diagram, the functional criteria, etc. ¨Future architects were primarily responsible for the project's entire technical and drawing aspects. For this purpose, they made inventories, created design assumptions, and developed all the visual elements so that the design was correct and attractive in terms of reception. It is important to emphasize that the architectural work also differed slightly depending on the specific design team. In the case of private investors, students had to work on an existing apartment, which they modified, adapting it to the needs of a given investor. In the group dealing with the project of the center for the Potrafię Pomóc Foundation, they created the concept of a new facility. Therefore, specific designs were different and focused on another problem of adapting a given environment to the needs of people with disabilities

- where is the evidence of this process? where is the data? exhibit the data of at least one case ¨Future occupational therapists conducted an in-depth interview based on the COMP questionnaire. They familiarized themselves with the dysfunction of a given patient/client and identified limitations and occupational problems. They analyzed the situation in an environmental and social context. They looked at the client's daily schedule, behavior patterns, tasks, and activities. As a result of analyzing the needs of a given investor, the therapists could signal to the rest of the team which part of the apartment area should be modified. They suggested introducing specific changes to the architectural plan and sometimes gave specific guidelines for designing tailored and individualized equipment supporting a given activity or occupation. In addition, they co-designed an ergonomic structure for a particular person with a disability

- the only exhibited figure of a design solution does not propose any novel solution for physical disability than what is already adopted as a standard in the majority of accessible institutions and private dwellings: ¨The conducted interview showed that the investor had difficulties carrying out daily hygiene activities and using the bathtub. Architecture students focused on this aspect and designed a bathroom tailored to the needs of a person with mobility difficulties. They proposed a general change in the room's space arrangement and equipped it with the necessary handles. They also suggested removing the bathtub in favor of a large shower equipped with a seat.¨ figure (IL.01). thus, it is not proven why interdisciplinary teams were needed or used in searching the accessibility design solutions? to reproduce the already existing accessibility standard code in design? it should have focused more on customized accessibility solution as the paper imply. 

- figures are not cited in the text. the authors must cite the figures in their corresponding paragraphs that describe them. 

- there is a severe lack of referencing various information throughout the manuscript, for example, ¨However, the most severe problem in this patient's case, limiting her independence, turned out to be dementia. This condition is characterized by the slow loss of cognitive function of the person suffering from it. Everything that surrounds her becomes foreign, and there is often a problem with naming objects and understanding their purpose¨

- this seems more like a secretary task, or graphical design not an interior design and architectural solution that would have utilized a more advanced spatial distribution to enable the fast recognition: ¨help the investor in everyday functioning and simultaneously enable her to exercise her memory and limit the aggravation of issues related to memory loss, the idea was to stick stickers on individual items and elements of interior design. This system was based on the investor's passion for solving crosswords and was developed by occupational therapists and a designer. Their development considered the colors, size, and font type. The students not only designed but also made these stickers and handed them over to their investor.¨ 

- the summary or conclusion does not exhibit at least the functional variation in the accessibility solutions for various types of disabilities at least in a table or a diagram. furthermore, how do we know that these solutions were functional and accessible, it is only mentioned interviewing the user in the manuscript but what are the questions in these interviews, what are the methods of quantifying the responses to the various questions, and where are the statistical analysis of these interviews. 

the manuscript needs major reforming in representing scientific data, especially in the experimental methods and results. Moreover, the manuscript must expand literature review over previous accessibility architectural projects with sufficient references. 

Author Response

Thank you for the review. 

We are sending you our manuscript. The article is changed, modified and corrected according as required by the review.

Round 2

Reviewer 2 Report

- The authors have performed some modifications on the manuscript as the added tables. however, more classification of the disabilities categories and questionnaire design and analyses of the users' satisfaction with the developed design in this case study research is still mandatory for presenting criteria for accessibility design in architecture not just to report a workshop case study. 

furthermore, the objective is not clear if it is around the pedagogical experiment of teaching architectural students through interactions and workshops about the accessibility design in architecture, which the reviewer believes is more relevant to the study scope and results, or if it is about generating accessibility design criteria for multiple categories of disability, or it is around customized accessibility design criteria. 

- the study still adopts a generalized voice and shallow methodology that touches only the surface of different topics (look at the previous comment). however in order to enhance the study and improve its impact, it is mandatory to focus on one of the three possible objectives listed in the previous comment and apply the necessary modifications corresponding to each case, for example:

- if the objective is customized accessibility design criteria (questionnaires of user satisfaction must be exhibited (figure for example) exhibiting how the questionnaire was designed and how the sample criteria were decided, and exhibiting statistical analysis of this questionnaire.

- or if the objective was a pedagogical approach for teaching accessibility design in architecture, evaluation criteria for the generated design must be exhibited, as well as the reached accessibility design criteria itself, etc.

the manuscript still requires this mandatory ordering in order to make it more clear and more significant.

Author Response

Thank you very much for your valuable comments, which allowed us to improve our work.

Response 1

We agree with the reviewer that the topic of universal design discussed in the work is not new in the literature. However, we believe that our proposal for joint education in the field of universal design and accessibility, students from various fields of study interested in this topic, as the first such initiative in Poland, deserves to be presented in a scientific journal. We wanted to draw attention to the need to create interdisciplinary teams in the implementation of housing projects, paying attention to the role of occupational therapy practitioners. We believe that the proposed program of training and workshops with the participation of investors (people with disabilities) is innovative and contains all the postulates recommended by the WHO.

Response 2

We reviewed the literature on academic education programs implemented in multidisciplinary teams focused on issues related to creating affordable buildings and apartments per the principle of universal design (UD). This remark has been taken into account in the introductory chapter, which we have reorganized and added two subsections. "The role of occupational therapy in UD: The effort of teaching and learning"

 Response 3

The primary purpose of the manuscript was to show the experience of cooperation in multidisciplinary teams of students of technical and medical faculties in creating housing adaptation projects for people with various disabilities. We wanted to allow students of architecture to get acquainted directly with occupational problems and barriers accompanying people with disabilities in the living environment. We tried to adapt the living space to its user following the concept of "customer in the center" and "person-environment-occupation". For the purposes of this manuscript, two descriptive accounts of house modifications are provided. The disability of each person was classified according to ICF. Characteristics of a person with a disability, according to ICF, were presented by students of physiotherapy together with occupational therapy. If, in the reviewer's opinion, this description is necessary for this paper, it will be included.

Response 4

In response to the comment regarding the accidental description of the experiment methodology, we have included Table 1 with the JAI project schedule, in which we show the experiment's stages and the participants' involvement. In Table 2, we have included the assessment of the main occupational problem of one of the participants (according to COMP). Some of the comments have been included in the following subsections of the work: 2.1. Inter-university Joint Architectural Initiative project (JAI) 2.2. The role of specific professions in the inter-academic project "JAI" 2.3. Participants/Disabled People/Investors.

            Regarding the extended workshop convention cited by the reviewer, we would like to present one of the reflections of an architecture student. This reflection was not included in the work: "Experiencing difficulties related to disability and becoming convinced of the existence of architectural barriers makes you extremely sensitive to the needs of others. Thanks to the fact that I had to face some of the challenges that people with disabilities face every day, I understood how important it is that every space, especially public space, is designed with all its users in mind…."

Response 5

The authors referred to the comments related to the visits. The description of the visitation has been arranged, and Table 1 has been added. These visits are intended to show the need for cross-sectoral cooperation between the economic sector and the academic community. In our work, we focused on the roles of individual team members during the implementation of the project and, in particular, on showing the importance of knowledge and competence in occupational therapy. Although we have detailed design documentation with solutions (3 proposals for each investor), we thought that including them would expand the work with architectural issues too much, which we wanted to avoid.

Response 6

In response to the reviewer's comments, the results of ADL, IADL, and the assessment of the difficulty of performing activities according to COPM of one of the participants (an elderly person with mild dementia) were added. In the case of a person with muscular dystrophy, such an analysis was not possible due to complete dependence on other people for self-service activities.

  Response 7

An innovative solution proposed to the investor/person with muscle atrophy was the design of an electric shaver, which was mentioned as the device that the patient dreamed of. This equipment was mentioned during the therapeutic interview prior to the start of the project. The patient's muscle strength was getting weaker and weaker every year, and he was less and less independent. And he desired to shave himself. That's why creative design students developed a shaver holder that was attached to the wall in the bathroom. Such a fixation method allowed the patient to drive the wheelchair himself, place his head in the right spot and be able to shave himself using his head and neck muscles. The sense of being in charge was essential for this patient since those were the only muscles he could still move on his own. This fact was crucial for the investor/person with muscle atrophy.

Response 8

Figure references have been added.

Response 9

This piece of text has been changed.

Response 10

In the given examples of solutions supporting the memory and functioning of the investor, her opinion and suggestions of the guardian were taken into account. The entire team accepted the final proposal.
